# Docosahexaenoic Acid Explains the Unexplained in Visual Transduction

**DOI:** 10.3390/e25111520

**Published:** 2023-11-06

**Authors:** Michael A. Crawford, Andrew J. Sinclair, Yiqun Wang, Walter F. Schmidt, C. Leigh Broadhurst, Simon C. Dyall, Larry Horn, J. Thomas Brenna, Mark R. Johnson

**Affiliations:** 1Institute of Brain Chemistry and Human Nutrition, Imperial College, London SW10 9NH, UK; yiqun.wang@imperial.ac.uk (Y.W.); mark.johnson@imperial.ac.uk (M.R.J.); 2Faculty of Health, Deakin University, Burwood, VIC 3125, Australia; andrew.sinclair@deakin.edu.au; 3Department of Nutrition, Dietetics and Food, Monash University, Notting Hill, VIC 3168, Australia; 4US Department of Agriculture, Agricultural Research Service, Beltsville, MD 20705, USA; walter.schmidt@usda.gov (W.F.S.); leigh.broadhurst@usda.gov (C.L.B.); 5School of Life and Health Sciences, University of Roehampton, London SW15 4JD, UK; simon.dyall@roehampton.ac.uk; 6MPEG LA, LLC, Bethesda, MD 20816, USA; horn.larry@gmail.com; 7Dell Pediatric Research Institute, Dell Medical School, Austin, TX 78723, USA; tbrenna@gmail.com

**Keywords:** docosahexaenoic, π-electrons, docosapentaenoic, di-DHA phosphatidylcholine, hexatriaconta-hexaenoic, retina, rhodopsin, non-classicality, vision, waveform, membrane, quantum-field, essential fatty acids

## Abstract

In George Wald’s Nobel Prize acceptance speech for “discoveries concerning the primary physiological and chemical visual processes in the eye”, he noted that events after the activation of rhodopsin *are too slow* to explain visual reception. Photoreceptor membrane phosphoglycerides contain near-saturation amounts of the omega-3 fatty acid docosahexaenoic acid (DHA). The visual response to a photon is a retinal *cis–trans* isomerization. The *trans*-state is lower in energy; hence, a quantum of energy is released equivalent to the sum of the photon and *cis–trans* difference. We hypothesize that DHA traps this energy, and the resulting hyperpolarization extracts the energized electron, which depolarizes the membrane and carries a function of the photon’s energy (wavelength) to the brain. There, it contributes to the creation of the vivid images of our world that we see in our consciousness. This proposed revision to the visual process provides an explanation for these previously unresolved issues around the speed of information transfer and the purity of conservation of a photon’s wavelength and supports observations of the unique and indispensable role of DHA in the visual process.

## 1. Introduction

Ever since the discovery of vitamin A by Elmer V. McCollum [1] in Wisconsin and its synthesis by Paul Karrer in Switzerland in 1937, there has been intense interest in its role in vision. In 1967, the Nobel Prize for Physiology or Medicine was awarded to Ragnar Granit, Haldan Keffer Hartline, and George Wald “for their discoveries concerning the primary physiological and chemical visual processes in the eye”. In his acceptance speech, Wald noted that “events after the activation of rhodopsin *are too slow* to explain visual reception” [2]. Hubel and Weisel later shared another Nobel Prize with David Sperry [3] in 1981. They showed how specific neurons were responsive to one eye by suturing one eye of a kitten. The current explanation for visual photo-transduction involves the absorption of a photon by 11-*cis* retinal, its isomerization to 11-*trans*, the breakdown of rhodopsin with the cascade of transducins, G-protein activation, and ion movements. This mechanism and the closing of the cGMP-gated ion channel create a negative potential with hyperpolarization at −65 mV. Its depolarization is considered the signal.

Yet, there are still four unknowns.

## 2. The Standard Model of Photo-Transduction

### 2.1. First, Speed of Response

Wald’s question on the speed of transduction can be illustrated by the capture of prey by the barn owl. When diving in darkness to capture a mouse rustling under leaves, the owl must process the sound differential information based on the difference in sound in the two ears, only about 10 cm apart, at a speed of 0.000015 s to locate and stay on track. The coordination of auditory, motor, and visual information demands a rapid processing speed if it is to extend its claws, grab the mouse, and change direction of flight in one sweeping movement. Such speed of processing requires electron function.

### 2.2. Second, Precision Is Required for the Signal but Is Not Explained by the Standard Model

If two photoreceptors respond differently to photons of the same wavelength, then visual acuity would be lost. Loss of visual acuity is commonly reported in experimental conditions where omega-3 fatty acid deficiency reduces the photoreceptor docosahexaenoic acid (DHA, all-*cis*-4,7,10,13,16,19-docosahexaenoic acid) content [4,5]. There is no sense of precision in the activation of cGMPs and transducins, nor in the ion movements.

### 2.3. Third, There Is No Explanation for the Transmission of the Vivid Detail Enabled by the Retina to the Brain for Reconstruction in Our Consciousness

How are images seen by the retina transferred in detail to the receptors in the brain? An attempt to explain the replication of the vivid detail seen by the retina in our consciousness proposed that photons were transferred [6]. However, Georgiev explained why this could not be [7]. Neither chemical transmitters nor ion movements explain the transfer. An electrical current per se would not carry specific information, such as wavelength. Yet, there is no explanation for the exquisite transfer of detail observed by the eye to our consciousness.

### 2.4. Fourth, How Is the Membrane Depolarized?

Membrane depolarization is generally considered to be the visual signal, but its mechanism is largely unknown.

## 3. Docosahexaenoic Acid (DHA)

The biosynthesis of the omega-3 family of polyunsaturated fatty acids (PUFAs) proceeds via a series of alternating position-specific desaturation and elongation steps from α-linolenic acid (all-*cis*-9,12,15-octadecatrienoic acid) to DHA or even longer hexaenoic acids [8]. However, DHA is preferentially incorporated into the brain during growth by an order of magnitude greater than biosynthesis [9], and is selectively incorporated into synapses, mitochondria, and microsomes compared to other omega-3 PUFAs [10].

DHA is densely packed in the membrane surrounding rhodopsin [11], yet its contribution to the signaling process has seldom been considered, even though it was shown in 1973 to be involved in the electrical function of the photoreceptors [12]. Moreover, DHA deficiency has been reported to adversely affect visual acuity in infants [4] and rhesus monkeys [13]. Klaus Gawrisch and colleagues have detailed how the DHA physical properties in the photoreceptor membrane differ from the omega-6 fatty acid all-*cis*-4,7,10,13,16-docosapentaenoic acid (DPAn-6), but not from DHA’s immediate precursor, the omega-3 fatty acid all-*cis*-7,10,13,16,19-docosapentaenoic acid (DPAn-3) [14,15,16]. This is a puzzle, as the precursor would originally have been more easily synthesized since the final desaturation is rate-limited and energy-costly. So why is the last double bond so critical and DHA so highly conserved in photoreceptors, neurons, and synapses for over 500 million years of animal evolution?

Bazan and Gordon published radio-autograph evidence of co-migration of rhodopsin and DHA during the reconstruction of the phagocytized material in the pigment epithelium for the rod, where the data indicated non-covalent binding of DHA to rhodopsin [17]. This evidence of a connection between DHA and rhodopsin is further supported by biophysical analysis from Gawrisch’s group, illustrating phosphatidylethanolamines accumulating preferentially near the protein, and these phosphoglycerides are particularly enriched in DHA. Consequently, there is growing evidence for the role of weakly specific DHA-rhodopsin interactions [15,16].

In the following sections, we will present a hypothesis whereby a DHA electron absorbs the surplus energy of the photon-induced isomerization, depolarizes the membrane, and transfers the information of the energizing photon to the brain, thereby resolving some of the unanswered questions and unknowns found with the standard model of photo-transduction.

## 4. Energetic Considerations

The law of conservation of energy dictates that the surplus energy from the photon energization of the electron and the *cis–trans* difference in energy must go somewhere and cannot be lost. The surplus energy from the isomerization is the sum of the energy of the photon (its wavelength) plus the difference between *cis* and *trans* retinal. The sum of the two functions would be at a higher energy than the exciting photon. A higher energy means a shorter wavelength, which is where DHA absorbs light. However, we calculated that the *cis–trans* difference is 7.9 kJ/mole. The energy of green light at 550 nm is 2.25 eV. The energy at DHA maximum absorption (240 nm) is about 2.3 times greater at 5.17 eV. The energy from the photon-induced isomerization would therefore be insufficient to shift the wavelength into the UV range. So, we are left with the question: where does the surplus energy go?

The surplus energy of isomerization must be accounted for by dissipation as heat. If not removed, then would not the constant flood of billions of photons during daylight damage the retina? This surplus energy from a single excitation will be quantized and will include the quantum of the exciting photon. We suggest that the surplus energy is absorbed by a neighboring DHA with the excitation of a π-electron that is then extracted by the hyperpolarization, depolarizes the membrane, and is sent to the brain. An artistic impression of the excitation of DHA in the photoreceptor membrane is shown in Figure 1.

## 5. Non-Classicality of Light Harvesting and DHA as an Energy Trap

The methylene group (-CH_2_-) interruption of the six double bonds found with the DHA molecule is essential to our thesis. It separates the π-electron clouds, creating an energy barrier and confining the electrons to their wells. Hence, DHA is an insulator, preventing the flow of a current across the membrane.

On the other hand, given an electrical potential across the DHA molecule that is sufficient to extract an electron, it will leave a hole that an electron with the same quantum mechanical properties can fill after tunneling through the energy barrier of the methylene groups. In that way, a current can flow. Thus, DHA acts as a semiconductor [18], with the excitation of a DHA electron lowering the barrier for conduction.

This idea is based on the extraordinary, high density of DHA in the phosphoglycerides, including DHA uniquely at both the *sn1* and *sn2* positions (di-DHA). Furthermore, in photoreceptor cells, the very long-chain omega-3 PUFAs are in the phosphatidylcholines (PC) of the outer segment membranes, tightly bound to rhodopsin [19]. Indeed, these retinal very long-chain hexaenoic fatty acids are mainly esterified at the *sn1* position of PC, with DHA in the *sn2* position [8]. These very long-chain PUFAs are only synthesized in the retina and testes [20]. It has been shown that loss or reduced levels of these very long-chain omega-3 PUFAs may cause loss of photoreceptors or functional perturbations [8,21]. Prominent in this very long-chain omega-3 PUFA family is the C36:6n-3, hexatriacontahexaenoic acid (HTA) [8,19,22], which places its omega-3 hexaenoic sequence between the outer and inner leaflets of the photoreceptor membrane. The location of the hexaenoic motif from HTA creates a stunning image of a high-density electron profile across this signaling membrane. DHA is present in the photoreceptor membrane phosphoglycerides at approximately 50% of the membrane fatty acids. So, virtually every *sn2* position is occupied by a DHA in the outer and inner leaflets. Hence, there will nearly always be 12 π-electrons top and 12 bottom. An arrangement of DHA-HTA-DHA means there are three sets of π-electrons arranged as 12-12-12. If we consider a region of favored di-DHA packing top and bottom with co-existing HTA, the arrangement is 24-12-24 or 24-12-12 or 12-12-24. Even without HTA or other even longer chain PUFAs, there will be 24-24, 24-12, or 12-24.

The outer leaflet, or membrane bilayer, is dominated by PC, and the inner by phosphatidylethanolamine. The phosphates cancel out, and the choline (-C-NH_4_^+^), a quaternary amine, is permanently and strongly positive, regardless of pH, with a pKb of 14.1. The ethanolamine of the inner layer is not ionized at physiological pH. Hence, there is always an intrinsic dipole moment across the membrane.

The planar structure of the energy-minimized DHA conformation is a fundamental characteristic of six double bonds separated by methylene groups. The π bonds always have a (+ve) end and a (−ve) end. The shape of their probability orbits will lean towards the (+ve) end of the intrinsic dipole moment. A simple mechanism for absorbing a quantized amount of energy is precisely that amount that will flip the direction of bond polarity to the opposite direction [23].

This electrical torsion from the inherent membrane dipole creates the image of the mass of DHA π-electrons under tension. Add HTA, placing its double bonds in the space between the bilayers, and there is a remarkable π-electron field density from top to middle and bottom of the membrane under the influence of this electrical torsion: energize an electron, and it will be like a hair trigger. In the presence of hyperpolarization, it would be extracted, and the electron could tunnel from top to bottom, de-polarizing the membrane. Interestingly, depolarization is considered to be the signal, although the mechanism is poorly understood.

The idea that there was a non-classical function at work in light harvesting was put forward by Wilde et al. in 2010 [24]. Energy transport has been reported to depend on the exciton–phonon interplay of molecular motions [25], and environmental fluctuations [26,27] with the formation of charge density waves at work [28]. Moreover, vibrations and quantum coherence can enhance energy transfer [29]. Evidence has been published showing that coupling to quantized vibrations is fundamental for biological function as this generates non-cascaded transport with a rapid and wider spatial distribution of excitation energy [30].

O’Reilly and Olaya-Castro have made the case for quantized vibrations in coherent light harvesting. They wrote: “we found that the properties of some of the chromophore vibrations that assist energy transfer during photosynthesis can never be described with classical laws, but this non-classical behavior enhances the efficiency of the energy transfer” [31].

## 6. Transfer to the Brain

Investigations into photo-transduction have not included any role for the DHA, electron-rich membrane structure surrounding rhodopsin and other opsins. Based on this non-classical behavior described above, it is plausible that a quantum field is responsible for energy transfer from the isomerization to the DHA surrounding rhodopsin. The idea of using quantum field theory in brain function is not new [32].

On the other hand, the idea is that the isomerization energy surplus is released as heat, which is vibrational energy that is quantized. With electronic/vibronic coherence already demonstrated, we have a novel and plausible mechanism for the quantized transfer of the isomerization energy to DHA.

The energy of the *cis–trans* isomerization will be common to all photoreceptions. The variable will be the energy of the photon (*h*c/λ). Hence, in each downstream event, we have a discreet, quantized packet of original information, namely the sum of the energy of the exciting photon plus that of the isomerization. This is the first step, as shown by (1).
E_DHAλ_ = Σ(*h*c/λ + E*_cis-trans_*)(1)
The energy pulse from a single isomerization will be quantized, and just as the photon ejected from a supernova retains the purity of its wavelength despite starting out several million light years away, this quantized energy cannot be lost. Thus, the electron extracted from DHA carries a quantized function of the original photon, offering a mechanism whereby the wavelength, which means color, is transferred intact to the brain.

Wavelengths are inversely related to energy E.
E = hf = hc/λ or λ = hc/E(2)
where h is Plank’s constant, c is the speed of light, f is the frequency, and λ is the wavelength.

Information could be carried by a single electron as a waveform and met by a single electron receptor. There is no explanation for how the color captured by the retina is transferred to our consciousness. The title of the 2007 paper by Rahnama et al. [33] expressed the question neatly: “How can the visual quantum information be transferred to the brain intact, collapsing there and causing consciousness?”. With the transfer by the actual photons themselves not being viable [7], the transfer by an electron must be considered.

The DHA electron energized by the isomerization surplus will be at an energy that contains a quantized function of the original photon that initiated the isomerization. The transportation of this single electron as a waveform could be via a tunnelling process or teleportation. It is analogous to a qubit. A flood of electrons will not work, as there is no discrete information. This idea of excess energy disposal is novel, as is the concept of DHA acting as an energy-electron transducer.

Support for this notion comes from direct evidence of quantum transportation [34], and the teleportation of photons using single photon receptors [35]. Photons and electrons have much in common. We are suggesting the presence of single electron detectors in the brain. There is, as far as we know, no other explanation for the transfer of imagery from the retina to the brain. The behavior of quantum systems in transferring information has been demonstrated by the Sycamore processor, which was shown to take 200 s to sample one instance of a quantum circuit, which would take a state-of-the-art classical supercomputer approximately 10,000 years to perform an equivalent task [36].

None of the above challenges the presently accepted model of events following photoreception, the formation of metarhodopsins, or the transducin cascade. It simply adds to the potential involvement of the high density of DHA. The specific, methylene-interrupted sequence of six double bonds in the membrane could act as a photon-energy-electron-transducer, providing a mechanism for quantized electrons to signal to the brain and enabling us to see a reconstruction of the vivid information collected by our retina.

It is plausible that the slower chemical reactions and ion movements on which Wald commented represent the charging of the battery as in membrane hyperpolarization leading to the extraction of an energized electron from DHA, with guidance provided en route to the brain. “Solving the human connectome at the nanoscale level of synaptic wiring will be critical for fully understanding the neural basis of human color vision” [37]. The novel hypothesis presented here requires further elaboration and testing. We have made a start in the next section with data supporting electron function and the potential of DHA for energy absorption [37].

## 7. DHA—Evidence for Electrical Activity

From molecular dynamic calculations, the ground state energy of DHA is 50 kJ/mol, which is in the region of energy reception of the order discussed here. We are proposing the absorption of the energy from the isomerization by DHA, with the hyperpolarization extracting the energized electron and depolarizing the membrane. This requires a condition in which DHA is electrically active. The fact that DHA is electrically active is supported by the first study of DHA deficiency in the photoreceptor performed on rats [12]. The deficiency did not affect structure but altered the electrical properties. The rod outer segment DHA concentration fell from 45.2% of total fatty acids to 19.0% in the DHA-deficient group. DHA was partially replaced with DPAn-6, which was the only major change identified between the groups. The rhodopsin concentration, the shape of the absorption spectra, and general bleaching characteristics were the same for the DHA-deficient and control groups. The density and packing of rods appeared normal in the DHA-deficient animals, and the ultrastructure of rod outer segments was indistinguishable from control. The electroretinogram (ERG) displayed a reduced A and B wave function. The A-wave of the ERG is a photoreceptor response function, while the B-wave is generated by electrical activity in other neural layers of the retina. Therefore, since these results indicate that specific decreases in membrane DHA lead to abnormal electrical activity, they are consistent with a unique and indispensable role for DHA in the electrical behavior of phototransduction.

Due to its six-methylene interrupted double bonds, DHA has a periodic structure. Any periodic structure on a quantum level can form energy bands and, under the right conditions, conduct electrons. We have previously provided evidence to support the potential of DHA being electrically active by using the calculation of the Kronig–Penney model, which showed that an energy band can exist in DHA [18]. We also tested the possibility of π-electron behavior using the nuclear Overhauser effect (NOE), which demonstrated polarization buildup at two ethene (CH=CH) sites at 127.6 ppm and at 132.8 ppm, the phosphatidyl, and, surprisingly, the methyl group [18].

If an energized electron from DHA is followed by a flow of electrons in the membrane depolarization, we are concerned here with this specific electron, which is at a higher energy. According to the law of conservation of energy, the wave function of that electron should carry a quantum of energy belonging to the original photon, thus differentiating it from the pack. It will be traveling at a speed that would certainly be fast enough for transduction, as it could “jump” the synaptic connections in the retina and lateral geniculate nucleus (LGN), rather than having to undergo the slower process of chemical nerve transmission. All that is required is that it be identified by single electron detectors.

For electrons specifically, individual spins in semiconductor quantum dots have been shown to stand out for their long coherence times and potential for scalable fabrication [35]. Moreover, pulse electron paramagnetic resonance spectroscopy has been used to show that the spin state of an electron can be teleported with high fidelity [38].

It is known that brain function is dependent on electron movement. If there is none, the brain is dead. In considering the options to explain the purity of the transfer from the retina to the brain, our suggestion here is that the transfer is by an energized electron from DHA as a wave function, which will include the energy quantum of the photon. That is the signal that provides pure information about color from the retina to the brain. Under the condition for DHA to conduct, a flood of or even a burst of electrons would not have the specificity of the signal but could provide a carrier wave to transfer the signal. On the other hand, it is also plausible that the transfer of a single, energized electron to the reception site in the brain could be by teleportation, electron tunnelling, or some other, not yet described, mechanism.

Our understanding of quantum behavior in the brain is in its infancy. The key issue is the question of speed and, secondly, the precision of the wavelength or energy quantum, to enable us to see in color. Both are answered by the transfer of an electron wave function carrying the information of the photon, as described above. Both are absolute requirements for color vision.

## 8. Raman Spectroscopy and Molecular Dynamics

Our data from Raman spectroscopy (see [39]) supports the idea that DHA could respond to vibrational energy and helps differentiate DHA from DPAn-3, shown in Figure 2 and Figure 3. The Raman data is evidence of the vibrational behavior of the hexaenoic acid sequence, which is of a different order than that observed in DPAn-3. In any =C-CH_2_-C= moiety, unequal =C-C and C-C= lengths result in twisting at the methylene CH_2_.

Curvature occurs when the lowest-energy H spin states in the C-H bond are −1/2 and +1/2 (or +1/2 and −1/2), resulting in twisting at the methylene site. Curvature will also result at the methylene site’s carbon atom due to electron spin states. C-C bond lengths will be unequal in C-C= and =C-C when electron spin states in the C-C bond are −1/2 and −1/2 versus +1/2 and +1/2. The spin sets −1/2 and +1/2 (or +1/2 and −1/2) would have an intermediate average bond length.

A change in curvature can be labeled as a change in carbon backbone bending. The accumulation of twisting and bending site-to-site results in the planar C shape of DHA, where three double bonds are planar. This is not possible with DPAn-3 [18].

Raman spectroscopy provides unequivocal imagery of the utility of DHA’s six-double-bond sequence compared to its five-double-bond precursor, DPAn-3. It also supports the concept of non-classical energy transmission. In addition, the scissoring and twisting on the DPAn-3 molecule with an odd number of double bonds implies it is less of a candidate for wave absorption.

Overall, the data presented in Figure 2 and Figure 3 suggest that DHA is a resonant molecule that has all it needs to be positioned at the receiving end of the vibrational energy generated by a photon-induced isomerization event.

### Di-DHA PC

In Figure 4, the lowest-energy electron spin orbitals in DHA are compared with those of di-DHA PC present in the photoreceptor. The results show that spin orbital energy levels in each of the two DHA moieties in the PC structure remain similar to each other, and both are close to the lowest electron spin orbital in DHA alone. The near absence of electron spin density between 0 eV and −2 eV is common. The energy levels for positive electron spin orbitals from 0 eV to +2 eV (pink) are about 10–20% broader in di-DHA PC than in DHA. Interestingly, DPAn-3 did not exhibit such clear differences, further emphasizing the unique nature of DHA.

In Figure 5, the assignment of positive electron spin orbitals to methine H sites holds for di-DHA PC. The lowest electron spin orbitals include those for carbon atoms as well as H atoms [40]. The broadening of the negative electron spin orbitals suggests the electron spin states for the carbon atoms in the methine sites again differentiate DHA from di-DHA PC molecular properties. The presence of di-DHA phosphoglycerides in the photo-receptor/rod/cone membranes likely enhances the probability of energy capture.

## 9. Summary and Conclusions: DHA as a Photon-Energy-Electron Transducer

In this paper, we have set out the limitations of the current model of photo-transduction and provided experimental evidence to support a previously unrecognized key role for DHA in the process. In this revision to the process, the quantum of energy from the photon-induced *cis–trans* isomerization of retinal is absorbed by a DHA π-electron. Hyperpolarization extracts the energized electron, which then transmits its information to the brain, conserving the fidelity of the original wavelength. This hypothesis provides a clear rationale to support the observations of the extraordinary high density of DHA and the presence of very long-chain omega-3 PUFAs surrounding the opsins, which serve to absorb the energy, and DHA acts as the photon-energy transducer sending visual information to the brain for image reconstruction in our consciousness.

In so doing, this thesis provides an explanation for the previously unresolved issues around the speed of information transfer and the purity of the conservation of a photon’s wavelength. It also explains the extraordinary concentration of the hexaenoic motive in the lipids adjacent to and attached to rhodopsin. This relationship has been conserved throughout the evolution of vertebrate species [41]. Our proposal that DHA acts as a photon-electron transducer for visual reception places DHA in an unassailable position at the center of visual signaling, and on a par with vitamin A. 

## Figures and Tables

**Figure 1 entropy-25-01520-f001:**
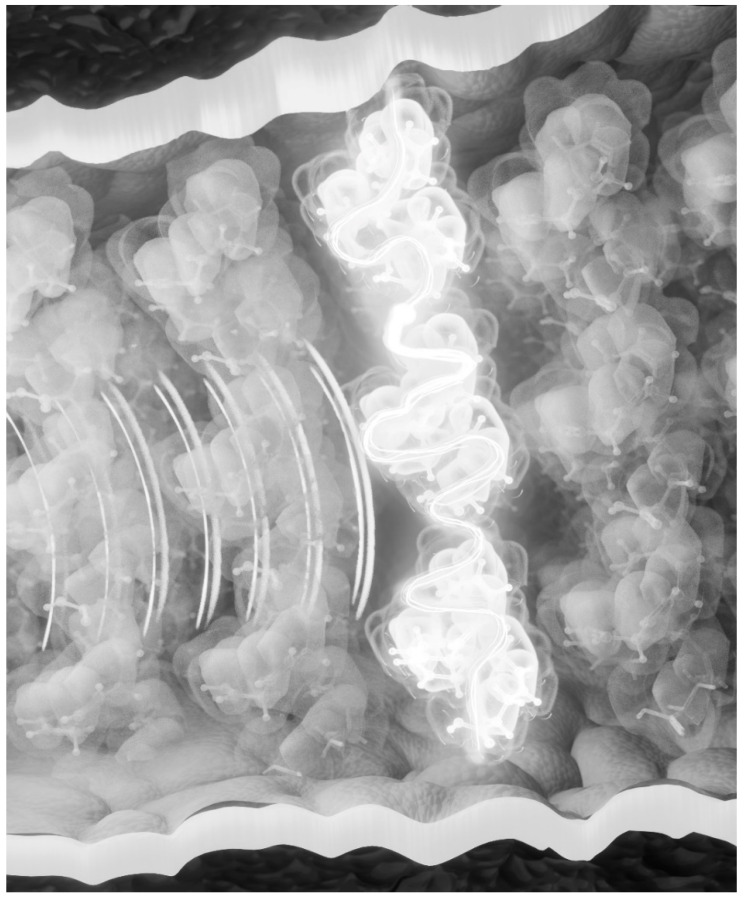
An artist’s impression of the excitation of DHA in the photoreceptor membrane (Kevin Heyse, Sage Marketing Group, Inc., Fort Collins, CO, USA). DHA absorbs surplus energy from the isomerization, which leads to electron excitation and its escape with membrane depolarization.

**Figure 2 entropy-25-01520-f002:**
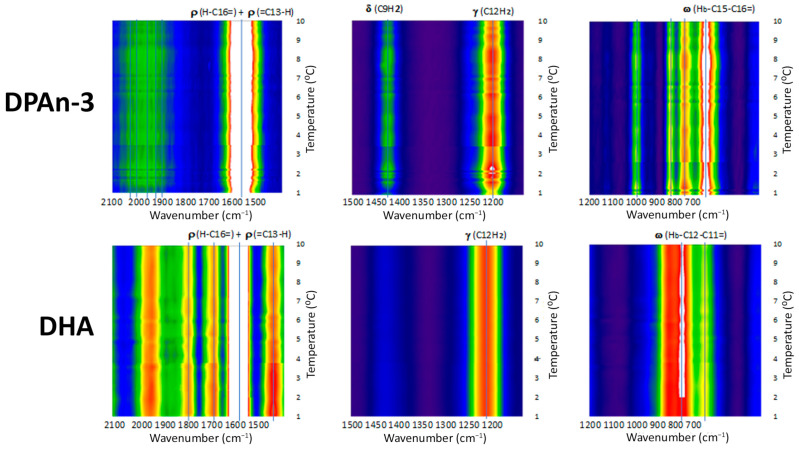
Gradient temperature Raman spectroscopy data for DHA and DPAn-3. Colors range from white (highest signal intensity) to red, orange, yellow, green, blue, to dark blue (lowest signal intensity. The red and green represent δ-positive and δ-negatives. There is a striking contrast between DHA and DPAn-3, especially in the high-frequency vibrational modes (2100-1500). These data suggest a high probability that DHA will accept vibronic information, with the energy being absorbed by an outer orbital electron. In this wavenumber region, the only common feature between DHA and DPAn-3 is white, which refers to the C=O. Adapted from [39].

**Figure 3 entropy-25-01520-f003:**
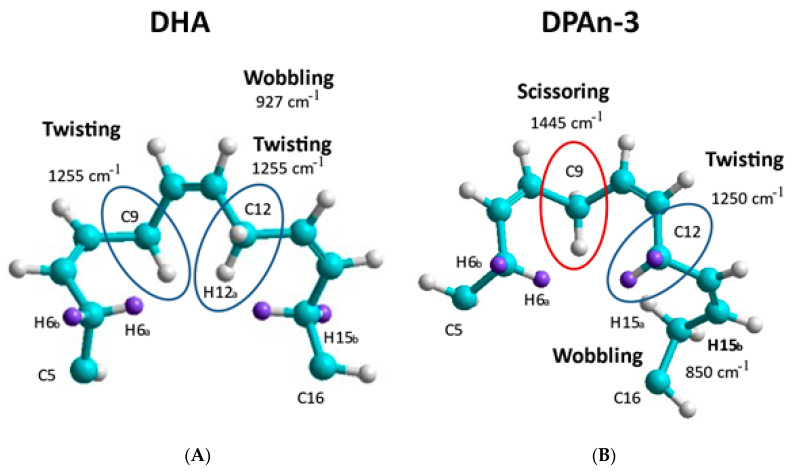
Vibrational modes generated by the computation chemistry of DHA and DPAn-3 are compared. (**A**) DHA, sites C5-C16, and (**B**) DPAn-3, sites C5-C16. Spontaneous shape-change behaviors are illustrated by comparing sections of the two molecules. Additionally, with DHA, three double bonds can be planar, whereas with DPAn-3, there are only two. The greater planarity is also consistent with the potential for the absorption of vibrational energy. Light blue atoms are carbon, white atoms are hydrogen. The purple atoms are also hydrogen highlighted to show in DHA twist at C5 is very different from C5 in DPAn-3. Adapted from [39].

**Figure 4 entropy-25-01520-f004:**
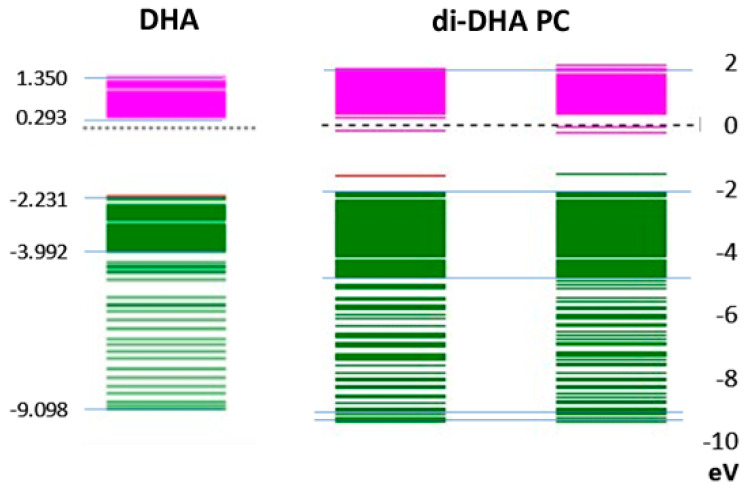
DHA and di-DHA PC spin orbitals are compared. There is a clear distinction between the molecular properties of DHA and di-DHA PC. The broadening from −2 to −4 to −5 ev and at −9 to −10 ev in the di-DHA is particularly noticeable, so much so that it raises the question as to whether it is specifically engaged in the energy transductions as it will likely be more sensitive than the single DHA. Moreover, the left-hand arm has more broadening from −6 to −9 ev, which could be significant. The green color is for spin orbital below the molecular plane. The electrons on protons below the plane are forced closer together due to the repeating *cis*-bonds. The redundancy in electron spin orbital methylene site to methylene site are similar but not identical. The pink color corresponds to electron spins at sites above the molecular plane.

**Figure 5 entropy-25-01520-f005:**
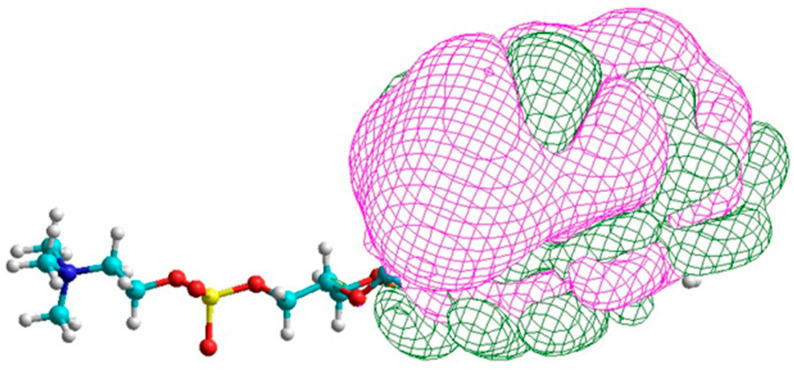
Di-DHA charge density around the hexaenoic motive, calculated by Hyperchem version 8. This molecular dynamic model of di-DHA supports the idea that di-DHA has a specific role in energy transduction. The light blue atoms are carbon, the white atoms are hydrogen, the red atoms are oxygen, the dark blue atom is nitrogen, and the yellow atom is phosphorus. The complexity of the total charge density, however, clearly rests in the *cis*-double bond region containing two DHA moieties. The orbital positive charge density is labeled pink, negative is labeled green. Pi bonding total charge density above versus below the molecular plane are not perfectly symmetrical.

## Data Availability

No new data were created or analyzed in this study. Data sharing is not applicable to this article.

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
