# Peer review of "Docosahexaenoic Acid Explains the Unexplained in Visual Transduction"

_entropy, 2023, doi:10.3390/e25111520_

Round 1

Reviewer 1 Report

Comments and Suggestions for Authors

This paper is potentially very important, but it needs a thorough rewrite.  There is no doubt that Wald's worry about the slow speed of the classical Hartline photoreceptor hyperpolarisation response is a serious problem, very graphically depicted in the barnowl story in the introduction. If I understand the hypothesis aright (and it needs much better description), DHA embraces the photosensitive opsin, and actually acts as an active multi welled semiconductor.  So it can initiate at the cis trans absorbtion of a photon by the opsin, a  quantal change of state that could set up a depolarising wave travelling from well to well at the  speed of light, even into the cerebral cortex, thus allowing the owl the microseconds needed to allow it to adjust its dive when the mouse first sees it coming. Even this description has required a five line sentence which will make the average reader despair.  This is the main problem with the paper.  It consists of short assertive sentences usually leaving out a host of essential background information and it needs thorough revision to correct this.  

I am an electrophysiologist and not an expert quantal physical chemist, so I have to assume the essential physical chemistry part of the story is correct.  But at times the text reads very naively - as if the authors are not aware of the fact that photoreceptors hyperpolarise after they absorb a photon, or that conscious sensation occurs immediately a signal reaches the visual cortex.  The narrative needs much better linking of the ideas with necessary background information, and the figures need much better explanation and linkage with the narrative.

Comments on the Quality of English Language

See above

Author Response

Thank you for your suggestions. Attached please find our response to your comments

Reviewer 2 Report

Comments and Suggestions for Authors

Dear authors!

I think this manuscript is of very good quality and can be published after minor changes.

Please standardize the naming of fatty acids and carry them logically throughout the manuscript in the formats "n-x HHH" or " ωx HHH", do not mix nomenclature.

Please replace PUFAS with PUFAs in line 72

Please delete the "." before the question mark on line 88.

Please replace Kj/mol with kJ/mol in line 108.

Please replace Dha with DHA on line 119.

please replace "36 carbon ω3-hexaenoic acid (HTA)" with HTA on line 155, already inserted on line 133

Please correct the discrepancy between ω6 and the name of the fatty acid in line 235. If the fatty acid C22:5n-3 is meant here, it has already been introduced in line 84 anyway.

 In the third figure, please replace the formula (N-3)-DPA with "n-3 DPA".

In the fifth figure, "DI -DHA CHOLINE PHOSPHOGLYCERIDE" is given, please replace it with "DI-DHA PHOSPHATIDYLCHOLINE" as in the title of the subchapter.

Please delete the contents of line 348.

Please standardise the bibliography, e.g. in reference 3 full first names are used, references 20 and 22 start with ".", reference 22 has two "." at the end, double first names are in XY and X.Y. format.

Author Response

(The authors gave the same response as above.)

Round 2

Reviewer 1 Report

Comments and Suggestions for Authors

This paper is now good for  publlication